# Diagnostic Pathway and Clinical Significance of Premature Ventricular Beats (PVBs) in Trained Bicuspid Aortic Valve (BAV) Athletes

**DOI:** 10.3390/jfmk4040069

**Published:** 2019-10-15

**Authors:** Matteo Donadei, Lorenzo Casatori, Vittorio Bini, Giorgio Galanti, Laura Stefani

**Affiliations:** 1Sports Medicine Unit, Clinical and Experimental Department, University of Florence, 50139 Florence, Italy; matteo.donadei@gmail.com (M.D.); lorenzo.casatori@gmail.com (L.C.); giorgio.galanti@unifi.it (G.G.); 2Department of Medicine, University of Perugia, 06100 Perugia, Italy; vittorio.bini@unipg.it

**Keywords:** bicuspid aortic valve, premature ventricular beats, athletes

## Abstract

Purpose: Bicuspid aortic valve (BAV) represents a common congenital cardiac disease (1–2%) normally compatible with sports activity. In the case of competitive sports, eligibility can be barred by the presence of symptoms, aortic valve dysfunction, or arrhythmias. This investigation of a large cohort of BAV athletes aims to verify the prevalence of premature ventricular beats (PVBs) found in the exercise test (ET) at the first sports medicine clinical evaluation. Methods: A sample of 356 BAV athletes, regularly examined over a period of 10 years at the Sports Medicine Center of the University of Florence, was retrospectively evaluated for arrhythmic events found in the first sports medicine check-up carried out. The athletes (321 M and 79 F), aged between 8–50 years (mean age 21.8 ± 11.6), practised sports at high dynamic cardiovascular intensity (mainly soccer, basketball, and athletics). Criteria for participation included a 2D echocardiography and ET conducted at 85% of maximal effort. Ventricular arrhythmic events were reported if found to be ≥3 at rest and/or during the exercise test and for subjects with any other cardiac or systemic structural diseases. Individuals aged >50 were excluded from the study. The selected participants were matched with a control group of 400 athletes with similar levels of training (age 20.0 ± 9.9) without BAV. Results: Only 25 (7.02%) of BAV athletes showed PVBs at the ET. A total of 403 single PVBs and four monomorphic couples were observed; a polymorphic pattern was present in only three athletes, and only five had exercise-induced PVBs at peak. None had acute events or major arrhythmias. The difference in PVBs prevalence in BAV athletes vs. controls (PVBs 6.25%) was not significant (*p* > 0.05). Conclusions: The prevalence of PVBs is low in BAV athletes and appears not to differ from athletes without BAV. Despite this, the behaviour of PVBs at the ET should be considered for the major suspicion for arrhythmic events. More data in this field could optimize the cost/effectiveness ratio for eventual ECG Holter indications.


**Dear Editor,**


The letter concerns the prevalence of premature ventricular beats (PVBs) in athletes with bicuspid aortic valve (BAV) examined in Sports Medicine. BAV is a common congenital cardiac disease estimated to be present in 1–2% of the general population as well as in athletes [1]. Recently, there has been increasing interest regarding BAV disease as an aortopathy, focusing on structural modification [2]. Previous studies demonstrated that the different morphological patterns of BAV are not associated with an impairment of left ventricle function [3]. Despite the fact that BAV is often compatible with physical activity, further questions arise in the case of competitive sports, when there is evidence of PVBs in the ET [4]. It is well known that Sports Medicine pays particular attention to arrhythmic events occurring during ET in order to prevent the risk of sudden death [5].

The presence of PVBs is the most common indication for further in-depth analysis of athletes’ health, even in the absence of structural cardiomyopathy [6]. The interpretation of PVBs can represent a clinical dilemma, particularly in the context of Sports Medicine [7]. Studies on elite athletes have found that most ventricular arrhythmias occur in the absence of an underlying heart disease and tend to disappear when they’re not trained [8]. This suggests that PVBs may be considered a feature of the adaptivity of the athlete’s heart [9].

Therefore, we believe that the main criteria for disqualification from sport should be based on the analysis of the morphological characteristics and prevalence of exercise-induced PVBs at the ET, since PVBs or arrhythmias in BAV athletes have not been reported in the present literature. Despite this, Italian Sports Medicine guidelines for eligibility give the indications for the evaluation of BAV athletes, the eventual presence of PVBs at the ET, and also 24-hour ECG Holter monitoring [10]. No data are available in the literature about the effective prevalence of PVBs or arrhythmias in BAV athletes, with the exception of anecdotal cases [11], and no sufficient data support the eventual need to implement the clinical evaluation of this special category applying other tests. 

We are in the process of retrospectively investigating a large cohort of young BAV athletes with mild valve dysfunction who practise aerobic sports at high dynamic cardiovascular intensity [12]. The subjects have been submitted to the ET for eligibility and are compared to a control group similar in age and sport practised. 

This investigation aims to verify, in a large cohort of BAV athletes, yearly, and followed in a Sports Medicine Centre for at least 10 years, the prevalence of PVBs found in ET at the first sports medicine clinical evaluation.

The selection, up to date, has been conducted among 356 BAV athletes aged between 8 to 50 (mean age 21.8 ± 11.6), either male or female, who practised sports at high cardiovascular impact (predominantly soccer, basketball, and athletics). The purpose was, first of all, to match the data of the prevalence of PVBs with a control group of 400 similarly trained athletes (age 20.0 ± 9.9). At least 400 athletes, males and females, were randomly selected from our database with the same characteristics (except for BAV) and who practised sports at the same intensity. All subjects selected underwent echo diagnosis, performed in the same laboratory, confirming the presence of BAV; an ECG at rest; and an ET on a treadmill (following the Bruce protocol) conducted at least 85% of their theoretical maximal heart rate. PVBs were considered when they were in a number of ≥3 or in the presence of complex morphology, at rest basal ECG, and/or during the ET. Exclusion criteria included being aged over 50, the presence of cardiac symptoms, and any cardiac or systemic arrhythmogenic disease. 

## 1. Statistical Analysis

Data analysis was performed using SPSS ver. 25 (IBM). Median and interquartile range (IQR) was reported for continuous variables and analyzed by the Mann–Whitney U test. Categorical variables were expressed as frequency counts with their percentages. Statistical analysis of categorical variables was made through the Pearson χ2 test for independence. Statistical significance was established as *p* ≤ 0.05.

## 2. Results

Our results showed no substantial difference (*p* = 0.779) of PVBs prevalence in both BAV and controls (7.02% vs. 6.25%, respectively) when considering the entire study population. However, when considering only the subjects with PVBs, a higher frequency of PVBs was observed in BAV compared to controls (median 12, IQR 8–22 vs. median 8, IQR 4–15, respectively, *p* = 0.08). Despite this, investigation needs to be more deeply evaluated, especially for BAV athletes with evidence of PVBs, which is an important message besides the indication for ECG Holter monitoring. The need for a larger echocardiographic investigation to discover the eventual presence of BAV in athletes appears justified by the prevalence of the disease in the general population as well as in athletes [13]. Up until now, BAV has been investigated for potential aortic valve and aortic vessel damage [14]. Although BAV affects only 1–2% of the general population and is often compatible with physical activity [4], this research highlighted the need for an in-depth analysis for BAV screening in athletes with PVBs. To our knowledge, this is the first longitudinal study including a large cohort of subjects with BAV. Considering the non-substantial differences found, it could be reasonable to restrict the indication to Holter monitoring in the case of BAV and only for athletes with induced PVBs at the ET. The importance of promoting echocardiographic tests at least once in the life of an individual practising sports, especially at the initial phase of the sports activity, therefore seems to be relevant, especially to avoid missing a diagnosis of BAV. At present, this aspect appears not to be sufficiently considered in sports medicine evaluation, at least not at the first clinical check-up, and it is not highlighted in the current Sports Medicine guidelines.

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
