# Peer review of "Diagnostic Pathway and Clinical Significance of Premature Ventricular Beats (PVBs) in Trained Bicuspid Aortic Valve (BAV) Athletes"

_jfmk, 2019, doi:10.3390/jfmk4040069_

Round 1

Reviewer 1 Report

I like the idea of highlighting the fact that more in-depth examinations and pre-participation screenings should be done for athletes. However, I would like to see greater emphasis placed on the dangers of bicuspid aortic valve congenital disease and the missing detection of premature ventricular beats. As the manuscript stands now, it lacks significance and true purpose. Additionally, the low percentages shown in the results further display a lack of importance. With that being said, I do believe there is potential within the paper. If possible, with the limited word count, please add in more references to why we should be aware of PVBs in BAV athletes. 

Author Response

We want to thank the  reviewers  for  giving us  the  opportunity  to highlight some  aspect of the paper. All the suggestions  have been  accepted. The manuscript has been revised by a native  English teacher.

Rev 1 

Many thank  for  giving us  the possibility to improve  the manuscript. We hope  to have clarified  the aim of the  study and to have emphasized the  importance to detect , in sports medicine, the eventual PBVs in BAV in a specific range of amount ,  especially considering that BAV  is  the most common congenital disease : 2% of the  general population. Actually the guidelines do not specify  these aspects, especially   during  the  EMT. The references have been updated  particularly for the arrhytmmias guidelines.  

The paper has been  entirely revised by the intervention of native English teacher. Following your suggestions , the modifications requested have been rephrased  to give  to the  paper  more fluency 

Reviewer 2 Report

With the present letter, the authors aim at investigating the prevalence of PBVs in BAV Italian athletes and propose an implementation to the Italian Sports Medicine screening guidelines to possibly reduce the risk of cardiac event during sports practice. Although BAV affects only a small percentage of the overall population, this topic needs additional investigation in athletic populations.

The article needs major grammar, syntax, punctuation and spacing revisions.  I advise the authors to consider revision from a native speaker before resubmission.  Several sections along the manuscript are not clear; moreover, the research question and the aim of this letter are not clearly stated.

I advise the authors to make the following major revisions:

Consider revision of lines 38-39. Suggested wording: “The letter concerns the prevalence of premature ventricular beats (PBVs) in athletic individuals with bicuspid aortic valve (BAV).” Consider revision of lines 41-42. Suggested wording: “Although BAV disease is also referred as BAV syndrome and BAV aortopathy [2], previous studies demonstrated that the different morphological patterns of BAV are not associated to an impairment of the left ventricle function [3].” Add reference at the end of line 45 Consider moving lines 56-61 to line 47. This paragraph helps the reader understand the context in which the study was conducted and the problem that the authors are trying to solve. prevalence of PBVs” Add a description of the statistical analysis used Clearly state the inclusion and the exclusion criteria Report the technique used to identify the PBV (i.e. electrocardiogram, heart rate monitor, etc…) Report if subjects were tested in multiple locations or in one laboratory Consider revision of lines 80-81. Suggested wording: “Preliminary results showed no difference between BAV and Control (7.02% vs 6.25%, p=0.779) in the The following sentence is not clear, please rewrite: “On the contrary considering the subgroup of subjects with the presence of PBV only, is emerged a trend toward the significance for an increased frequency of PBV in BAV 83 (BAV: median 12, IQR 8-22 vs Controls: median 8, IQR 4-15, p=0.08).” Report a reference supporting the need of an echocardiographic exam (line 90-91) to identify BAV. The manuscript reports incomplete results, consider adding the results reported in the abstract. Conclusions in the manuscript and in the abstract are not in agreement.  Please review both results and conclusion paragraphs.

Other minor revisions:

Change “medium age” (line 70) with “mean age” Use “.” Instead of “,” to separate units from decimals “At least 400 athletes” (line 72) is too generic, report the exact number of participants Change “heart rate frequency” (line 77) to “heart rate”

Author Response

Many thank for help us  to  improve  the manuscript. Following your suggestions the paper has been in deep revised  by a native English teacher. All the majior and minor revision has been  accepted . We hope  to  have correctly modified  the text

Consider revision of lines 38-39. Suggested wording: “The letter concerns the prevalence of premature ventricular beats (PBVs) in athletic individuals with bicuspid aortic valve (BAV).”

The sentence has been  modified

 Consider revision of lines 41-42. Suggested wording: “Although BAV disease is also referred as BAV syndrome and BAV aortopathy [2], previous studies demonstrated that the different morphological patterns of BAV are not associated to an impairment of the left ventricle function [3].”

The sentence has been rewritten following your suggestions

Add reference at the end of line 45 Consider moving lines 56-61 to line 47. This paragraph helps the reader understand the context in which the study was conducted and the problem that the authors are trying to solve. prevalence of PBVs”

The modifications requested have been made

Add a description of the statistical analysis used

The statistical analysis description has been insert

Clearly state the inclusion and the exclusion criteria Report the technique used to identify the PBV (i.e. electrocardiogram, heart rate monitor, etc…) Report if subjects were tested in multiple locations or in one laboratory

 Thank you for giving us  the  opportunity  to clarify this  aspect . The  inclusion  and exclusion criteria  have been specified . The subjects  were investigated in the same  Laboratory

 Consider revision of lines 80-81. Suggested wording: “Preliminary results showed no difference between BAV and Control (7.02% vs 6.25%, p=0.779) in the The following sentence is not clear, please rewrite: “On the contrary considering the subgroup of subjects with the presence of PBV only, is emerged a trend toward the significance for an increased frequency of PBV in BAV 83 (BAV: median 12, IQR 8-22 vs Controls: median 8, IQR 4-15, p=0.08).”

 The  modfications have been made . We  hope  to  have correctly interpreted  the suggestions .

Report a reference supporting the need of an echocardiographic exam (line 90-91) to identify BAV.

A reference has been added

 The manuscript reports incomplete results, consider adding the results reported in the abstract.

As requested ,the results  reported in the  abstract are now in the text. We respectfully remember  that  the manuscript  has been  sent as a  letter , just  to clarify  the idea . However  considering now  , as  consequence  of your effort , and appreciation has been in deep modified we , kindly ask if is possible , in case  of acceptance to consider  an  other type , like as a short  article or similar.

 Conclusions in the manuscript and in the abstract are not in agreement.  Please review both results and conclusion paragraphs.

 Conclusions and results  have been modified

Other minor revisions:

Change “medium age” (line 70) with “mean age” Use “.” Instead of “,” to separate units from decimals “At least 400 athletes” (line 72) is too generic, report the exact number of participants Change “heart rate frequency” (line 77) to “heart rate” 

The manuscript has benne completely re-written with the support  of a native  English language teacher. Particularly  the spelling and  the  mistakes

Round 2

Reviewer 2 Report

The quality of the manuscript is improved.  The introduction is organic and methods report the most relevant information.  However, the manuscript needs important English language revision before it can be considered for publication.  Please have the manuscript reviewed by a native speaker before resubmission.  See additional comments below.

Format: Review spacing between words or between punctuations and words. Few sentences miss punctuation, please review.

Title: Remove abbreviations.

General comments:

Line 23: “… during the exercise test …”

Line 24: “… structural diseases. Individuals aged >50 were excluded…”

Line 24, 25: “The selected participants were matched with a control group of 400 athletes with similar level of training …”

Line 26, 27: “A total of 403 single PBVs and 4 monomorphic couple were observed;”

Line 72: “in order to verify some differences vs the control group” too generic, specify which difference you are looking at.

Line 73-75: Specify the number of participants in the BAV athletes even if you reported it in the abstract.   

Line 100-102: I advise the authors to first discuss the relevance of the topic studied in this research and then state what makes this article unique and different from the literature. For example “Although BAV affects only the 1-2% of the general population and it is often compatible with physical activity, this research highlighted the need of in depth analysis for BAV screening in athletes with PVBs.  To our knowledge this is the first longitudinal study including a large cohort of subjects with BAV.”

English language:

Line 44–46: “Recently there has been increasing interest regarding BAV disease and although it is also referred to as BAV syndrome and BAV aortopathy.”

Line 49–51: It is well known that sports medicine discipline pays particular attention to arrhythmic events occurring during EMT, 50 addressing the prevention of sudden death [5].

Line 52: The presence of PVBs is the most common indication for further in depth analysis of athletes’ health …”

The paragraph between line 58 and 60 reports a consideration of the authors.  Consider revision: “Therefore, we believe that the main criteria for disqualification from sport should be based on the analysis of the morphological characteristics and prevalence of exercise-induced PVBs at the EMT since PBVs or arrhythmias in BAV athletes has not been reported in present literature.”

Line 65 and line 95: change “necessity” with “need”

Line 75: “The main purpose of our investigation was to compare the data” specify which data you aim to compare “with a control group of 400 similarly trained athletes.”

Line 80-83: Sentence not clear, please review “PVBs were when they were ≥ 3 or in presence of complex morphology at rest basal ECG and/or during the Exclusion criteria include cardiac symptoms, being aged over 50 and the presence of any cardiac or systemic arrhythmogenic disease.”

Consider revision line 89-90: “Our results showed no substantial difference (p=0.779) of PVB prevalence in both BAV and Control (7.02% vs 6.25%, respectively) when considering the entire study population.”

Consider revision line 91-93: “However, when considering only the subjects with PBV, a higher frequency of PBVs was observed in BAV compared to Controls (median 12, IQR 8-22 vs median 8, IQR 4-15, respectively, p=0.08).”

Author Response

We want to thank  the  the  reviewer for the  details suggested and to  give us  the opportunity to  improve the manuscript especially  in terms  of main message . 

 All the  suggestions have been accepted and reported  in red  in the manuscript. 

 We hope  to have  correctly modified  the  text .